# Prevalence and associated factors of mental health disorders among Brazilian healthcare workers in times of the COVID-19 pandemic: A web-based cross-sectional study

Silvia Helena Mendonça de Moraes[1], Inara Pereira da Cunha[2], Everton Ferreira Lemos[3], Lesly Lidiane Ledezma Abastoflor[1,4], Maria de Lourdes Oshiro[2], Rosana Teresinha D. Orio de Athayde Bohrer[5], Vicente Sarubbi, Jr[3], Fabrícia Barros de Souza[5], Débora Dupas Gonçalves do Nascimento[1], Sandra Maria do Valle Leone de Oliveira[1]*

**1** Fiocruz Mato Grosso do Sul, Campo Grande, Mato Grosso do Sul, Brasil, **2** Escola de Saúde Pública Dr. Jorge David Nasser, Campo Grande, Mato Grosso do Sul, Brasil, **3** Universidade Estadual de Mato Grosso do Sul, Campo Grande, Mato Grosso do Sul, Brasil, **4** Universidade Federal de Mato Grosso do Sul, Campo Grande, Mato Grosso do Sul, Brasil, **5** Fiocruz Brasilia, Distrito Federal, Brasilia, Brasil

☯ These authors contributed equally to this work.
* sandra.leone@fiocruz.br

## Abstract

The COVID-19 pandemic in Brazil affected mental health among healthcare workers. To objective of this study was to evaluate the mental health of healthcare workers in in the central-west region of the Brazil, estimating the prevalence of mental health disorders, and investigating associated factors, perceptions of safety, and self-perceptions about mental health in times of the COVID-19 pandemic. The questionnaire was divided into two parts that included general information and perceptions about the work process and identified symptoms using the Depression Anxiety Stress Scale-21 (DASS-21), and multiple linear regression analysis was conducted. A total of 1,522 healthcare workers participated in the survey. Overall prevalence of symptoms was calculated for depression (58.7%), anxiety (59.7%), and stress (61.7%). Physicians had 3.75 times greater risk of depression (1.59–8.85, 95% CI). Independent variables associated with depression symptoms were not feeling safe with the way services were organized (1.12:1.03–1.21, 95% CI) and self-perception of poor mental health (8.06: 4.03–16.10% CI). Working in management was protective, and married professionals had 12% lower risk of exhibiting symptoms of depression (0.79–0.99, 95% CI). Participants with self-perception of poor mental health had 4.63 greater risk for symptoms of anxiety (2.58–8.31, 95% CI). Protective factors were not having sought support for mental health (0.90: 0.82–0.99, 95% CI), having a graduate degree (0.71: 0.54–0.94, 95% CI), and not having been diagnosed with COVID-19 (0.90: 0.83–0.98, 95% CI). Perception of poor mental health was associated with 6.95-fold greater chance of developing stress symptoms. Protective factors from stress were having a degree in dentistry (0.81: 0.68–0.97, 95% CI), residing in Mato Grosso do Sul (0.91: 0.85–0.98, 95% CI), and not having sought mental health support services (0.88: 0.82–0.95, 95% CI). The prevalence of mental health disorders is high among healthcare workers, and is associated with

**Data Availability Statement:** All relevant data are within the paper and its Supporting information files.

**Funding:** Programa Inova FIOCRUZ – Edital Geração de Conhecimento – Enfrentamento da Pandemia e Pós-pandemia Covid-19: Encomendas Estratégicas".

**Competing interests:** No authors have competing interests.

professional category, organization of services provided, and self-perception of poor mental health, reinforcing the need for preventative measures.

## Introduction

The COVID-19 pandemic and the implementation of related isolation measures led to an increase in mental health problems around the world, including depression and generalized anxiety disorder [1,2].

Globally, the impact of common mental disorders among healthcare workers is underestimated [1]. Prior to the arrival of COVID-19, attention to the health of workers (including mental health) had never been a priority in healthcare policy. Understanding that health professionals are on the front line and need to be protected to ensure better health for everyone, as emphasized in the 2030 Sustainable Development Goals, remains an ongoing challenge, especially in countries where resources are limited [3].

Healthcare workers as a group should be considered vulnerable to sickness and even mortality during health crises. Besides biological risk, their mental health is more likely to be affected compared to the general population [4–6]. An estimated one fourth of the healthcare workforce exposed to COVID-19 developed anxiety, depression, acute stress, insomnia, posttraumatic stress symptoms, and burnout [7,8].

Fear, social distancing, and continuous feelings of anguish and concern were observed throughout society during the pandemic [9–11]; healthcare professions, however, involve specific challenges [12–14] such as poor working conditions and remuneration [7], insufficient and inadequate patient beds [15], poor quality and insufficient quantities of individual protective equipment [16,17], as well as longer working periods and consequently fewer rest hours [18]. Unusual situations involving moral suffering and other ongoing dilemmas inherent to routine activities in healthcare that were noted during the COVID-19 pandemic have also been described [19,20].

The mental illness of healthcare workers is a complex and multifaceted phenomenon, influenced by individual, interpersonal, organizational, and social factors [21,22]. To protect the mental health of these professionals, it is important to implement protective measures, such as organizational support, effective communication, adequate training, and access to mental health resources, to mitigate risks and promote the resilience of healthcare workers in times of health crisis [23].

Despite the investigation of psychosocial risks and protective measures for the mental health of healthcare workers during the COVID-19 pandemic [21–23], further research is still needed. New virus variants are emerging and the pandemic is still evolving, which may have a different impact on the mental health of healthcare professionals [24]. In addition, working conditions for healthcare professionals can vary significantly across different countries and regions, and the support and protection measures offered to workers can also vary [25].

Therefore, it is important to continue studying the mental health of healthcare workers to understand the specific needs of professionals in different contexts and to develop tailored interventions that can meet their needs [26].

The objective of this study was to evaluate the mental health of healthcare workers in Brazil, estimating the prevalence of mental health disorders, and investigating associated factors, perceptions of safety, and self-perceptions about mental health in times of the COVID-19 pandemic.

## Methodology

### Study type

This web-based cross-sectional study was conducted from November 2020 to October 2021, according to the STROBE recommendations [27].

### Selection and sample

Initially, we obtained the consent of each professional class boards, in order to carry out invitations to professionals and obtain active professional records for validation of research data. After approval by the Research Ethics Committee, the invitations initially came from the class boards.

Only medical professionals, nurses, nurse technicians, dentists, dentists techinicians, pharmacists, or physical therapists with active status with Brazilian professional boards were included.

In Brazil, there are health professionals with technical qualifications and high school education, recognized and registered in their professional bodies. To ensure the participation of only professionals, we requested in the Informed Consent Form (ICF), that they inform the Class Boards number, which was later validated with the active subscribers of each boards.

A total of 56,298 healthcare professionals were registered with their respective professional boards; the sample size was calculated based on the prevalence of mental health disorders among health professionals during the COVID-19 pandemic, using a rate of 34% and sample error of 5%, according to the following formula: $n = (z_{\alpha/2})^2 \, p(1-p)/e^2$, where $e = z_{\alpha/2} \sqrt{p\,(1-p)/n}$ [28]. This yielded an estimated total of 1,280 participants, considering 10% sample loss. The non-probabilistic sampling was proportionally distributed between Mato Grosso do Sul and the Federal District and among the various professions.

### Data collect

An electronic form on the REDCap platform was used to collect the data from voluntary respondents, healthcare workers from two states in the center-west region of Brazil.

The participants responded to an online questionnaire which included questions about sociodemographic aspects related to work and to health. The questionnaire was divided into two parts that included general information and perceptions about the work process.

The following strategies were used to invite participants: email sent to the professional by the profession's class board and dissemination on social networks with the research link.

Data entry into the RedCap system was monitored by research managers, who validated the information. Managers accessed the system using a login and password, keeping the participants' sensitive data confidential.

### Symptoms mental disorders—validated scale

Three outcomes for mental health disorders were considered (symptoms of depression, anxiety. and stress) using the Depression Anxiety Stress Scale-21 (DASS-21), adapted and validated for Portuguese [29]. The DASS-21 is a self-reported assessment containing three subscales graded according to a 4-point Likert scale (0–3, with 0 corresponding to "Disagree completely" and 3 "Agree completely"). Each subscale of the DASS comprises seven items that evaluate the emotional states of depression, anxiety, and stress.

## Data analysis

For the multivariate analysis, only the variables that demonstrated association in univariate analysis were included.

For the dependent variable depression, data were collected on the following: school completion, professional category, physical health classification, mental health classification, COVID-19 diagnosis, level of safety in facing COVID, safety in work organization, leave, psychological/psychiatric follow-up before the pandemic, psychological/psychiatric follow-up during the pandemic, marital status, presence of a partner, occupation, workload, distancing, reallocation, how the pandemic affected income, and state.

For the dependent variable anxiety, the data collected included: sex, school completion, professional category, physical health classification, mental health classification, COVID-19 diagnosis, safety in facing COVID, safety in work organization, leave, psychological/psychiatric follow-up before the pandemic, psychological/psychiatric follow-up during the pandemic, living companions, workload, main work relationship, distancing, work situation, reallocation, and how the pandemic affected income.

For the dependent variable stress, data were collected on the following: sex, school completion, state, professional category, workload, physical health classification, mental health classification, COVID-19 diagnosis, safety in facing COVID, reallocation, leave, psychological/psychiatric follow-up before the pandemic, psychological/psychiatric follow-up during the pandemic, and how the pandemic affected income.

The data were analyzed using Stata SE software version SPSS 27.0 (StataCorp LP, College Station, USA). Associations were determined during univariate analysis using the chi-squared test ($\chi 2$) or Fisher's exact test (for categorical variables with expected frequency <5) to evaluate differences between proportions and determine 2-tailed p values. Variables with $p<0.20$ were included in linear Poisson regression, and prevalence ratios were calculated with robust adjustment of variance (RPaj) for each of the independent variables (symptoms of depression, anxiety, and stress).

Informed consent was obtained electronically written from all participants, with the approval of the Fiocruz Brasília Ethics Committee (#n. 4.401.333).

## Results

A total of 1522 healthcare workers in the center-west region were included, 45.4% in the state of Mato Grosso do Sul and 54.6% in the Federal District. The symptoms of mental health disorders found among the healthcare workers were depression (58.7%), anxiety (59.7%), and stress (61.7%), as shown in Fig 1.

The respondents were mostly women (82.6%), self-declared as white (52.7%), and lived with a spouse (45.1%). Nearly 25% of respondents who completed higher education reported the presence of depression or anxiety. Respondents in the Federal District reported higher rates of depression (57.7%), anxiety (56.2%), and stress (57.4%) (Table 1).

Professionals working in nursing (nurses and nursing technicians) reported higher frequencies of mental disorders; depression, anxiety, and stress were present in approximately 32% of nurses. As for professional credentials, participants with COREN (regional nursing board registration) and CRM (regional board of medicine registration) presented higher rates for the three outcomes (Table 2).

A high percentage of respondents with depression, anxiety, and stress considered their physical and mental disposition for personal and professional demands during the pandemic to be moderate, varying from 44.30% to 49.40%. Most respondents with depression, anxiety, and stress were diagnosed with COVID-19 (64.22%, 63.30%, and 64.82%, respectively)

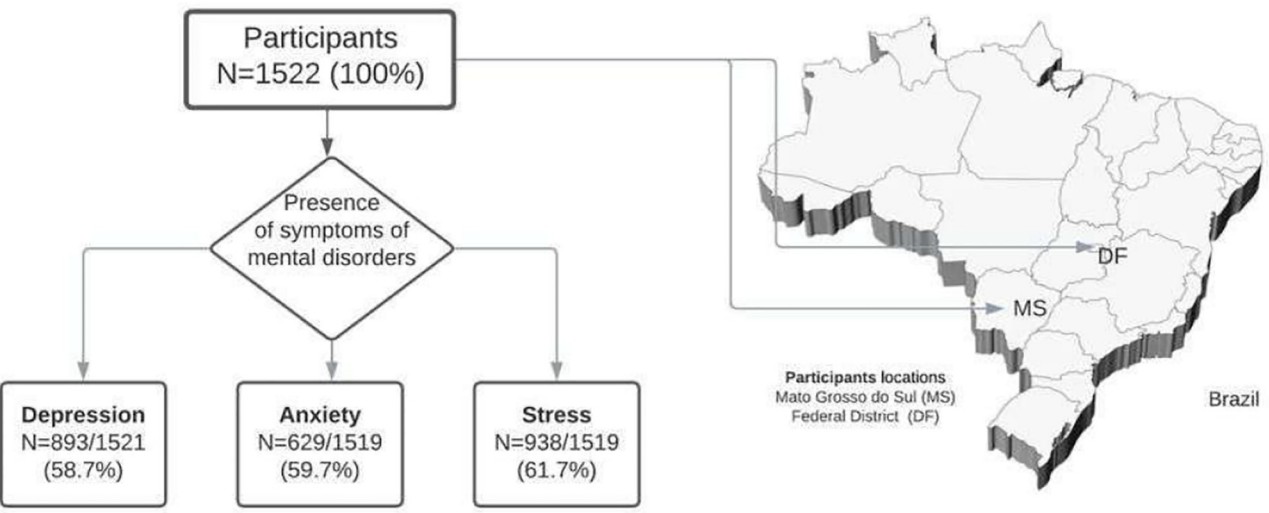

**Fig 1. Flowchart of participants with symptoms of mental disorder according to the location of residence.**

maintained social distancing outside of work (93.82%, 93.96%, and 93.72%). The majority also stated that they did not receive psychological or psychiatric care prior to the pandemic (71.44%, 71.08%, and 70.33%) (Table 2).

The significant results of multivariate analysis using Poisson regression can be found in Table 3.

Physicians were 3.75 times more likely to experience symptoms of depression than the other professional categories (95% CI 1.59; 8.85). Additionally, not feeling safe with regard to the organization and structure of one's work in the face of the pandemic was associated with depression (RPaj: 1.12, 95% CI 1.03; 1.21). Furthermore, married healthcare workers had a 12% lower rate of depression than other marital status categories (RPaj: 0.88, 95% CI 0.79; 0.99). Employment in the areas of management, nursing, and pharmacy were also considered to have a protective effect (p<0.005).

The perception of poor mental health was associated with a 4.63-fold risk for symptoms of anxiety. Factors associated with risk were specialization (RPaj: 0.71, 95% CI 0.54; 0.94) or master's degree (RPaj: 0.70, 95% CI 0.51; 0.95). Protective factors were not having been diagnosed with COVID-19 (RPaj: 0.90, 95% CI 0.83; 0.98) and not having sought psychological and/or psychiatric help or treatment during the pandemic (RPaj: 0.90, 95% CI 0.82; 0.99).

Factors associated with stress were self-assessment of mental health as moderate or poor. These variables had 6.1 to 6.9 times the risk for stress. Having a degree in dentistry (Rpaj: 0.81–95% CI-0.68; 0.97), living in the state of Mato Grosso do Sul (Rpaj:0.91, 95% CI-0.85; 0.98), and not having sought psychological and/or psychiatric help or treatment during the pandemic period (Rpaj: 0.88–0.82; 0.95) were protective factors against signs and symptoms of stress.

## Discussion

This study found that over half of healthcare workers surveyed reported some type of mental health disorder within the context of the COVID-19 pandemic in Brazil.

In Latin America during the early part of the pandemic, signs and symptoms of these disorders were estimated to be high: prevalences of 37% for anxiety, 34% for depression, and 33%

**Table 1. Descriptive analysis and association of variables related to working conditions with outcomes (N = 1,522), MS/DF, Brazil, 2021.**

| Variable | N (%) | Depression | | | | Anxiety | | | | Stress | | | |
|---|---|---|---|---|---|---|---|---|---|---|---|---|---|
| | | Present N = 893 n (%) | Absent N = 629 n (%) | $X^2$ | p-value | Present N = 911 n (%) | Absent N = 611 n (%) | $X^2$ | p-value | Present N = 941 n (%) | Absent N = 581 n (%) | $X^2$ | p-value |
| **Occupation (professional activity)** | | | | | | | | | | | | | |
| Nurse | 488 (32.1) | 291 (32.6%) | 197 (31.2%) | 5.30 | **0.021** | 300 (32.9%) | 188 (30.9%) | 1 | 0.138 | 307 (32.6%) | 181 (31.2%) | 28.77 | **0.001** |
| Nursing technician | 363 (23.9) | 253 (28.3%) | 110 (17.6%) | | | 256 (28.1%) | 107 (17.5%) | | | 247 (26.2%) | 116 (20.0%) | | |
| Physician | 198 (13.0) | 112 (12.5%) | 86 (13.7%) | | | 101 (11.1%) | 97 (15.9%) | | | 118 (12.5%) | 80 (13.9%) | | |
| Dentist/Oral surgeon | 171 (11.3) | 69 (7.7%) | 102 (16.2%) | | | 78 (8.7%) | 93 (15.2%) | | | 82 (8.8%) | 89 (15.3%) | | |
| Pharmacist | 101 (6.6) | 62 (6.9%) | 39 (6.2%) | | | 64 (7.0%) | 37 (6.1%) | | | 66 (7.0%) | 35 (6.0%) | | |
| Physical therapist | 95 (6.3) | 49 (5.5%) | 46 (7.3%) | | | 54 (6.0%) | 41 (6.7%) | | | 64 (6.8%) | 31 (5.3%) | | |
| Oral hygienist | 14 (0.9) | 7 (0.8%) | 7 (1.1%) | | | 8 (0.9%) | 6 (1.0%) | | | 9 (1.0%) | 5 (0.9%) | | |
| Management | 32 (2.1) | 17 (1.8%) | 15 (2.4%) | | | 20 (2.2%) | 12 (2.0%) | | | 21 (2.2%) | 11 (1.9%) | | |
| Educator | 13 (0.9) | 6 (0.7%) | 7 (1.1%) | | | 5 (0.6%) | 8 (1.3%) | | | 5 (0.5%) | 8 (1.4%) | | |
| Other | 44 (2.9) | 26 (2.9%) | 18 (2.9%) | | | 23 (2.5%) | 21 (3.4%) | | | 21 (2.2%) | 23 (4.0%) | | |
| **Professional category** | | | | | | | | | | | | | |
| COREN | 907 (59.6) | 573 (64.2%) | 334 (53.2%) | 19.39 | **0.001** | 587 (64.4%) | 320 (52.3%) | 23.68 | **<0.001** | 582 (61.8%) | 325 (55.9%) | 16.49 | **<0.001** |
| CRM | 205 (13.5) | 117 (13.1%) | 88 (14.0%) | | | 105 (11.2%) | 100 (11.0%) | | | 123 (13.1%) | 82 (14.1%) | | |
| CRO | 196 (12.9) | 80 (9.0%) | 116 (18.4%) | | | 90 (9.9%) | 160 (17.6%) | | | 93 (9.9%) | 103 (17.7%) | | |
| CRF | 115 (7.5) | 72 (8.1%) | 43 (6.8%) | | | 73 (8.0%) | 42 (4.6%) | | | 115 (12.2%) | 38 (6.5%) | | |
| CREFITO | 99 (6.5) | 51 (5.6%) | 48 (7.6%) | | | 56 (6.1%) | 43 (4.7%) | | | 66 (7.0%) | 33 (5.7%) | | |
| **Do you have more than one professional affiliation?** | | | | | | | | | | | | | |
| Yes | 464 (31.1) | 267 (29.8%) | 197 (31.3%) | | | 605 (66.4%) | 424 (70.2%) | 1.62 | 0.112 | 636 (67.6%) | 393 (67.6%) | 0.009 | 0.954 |
| No | 1029 (68.9) | 609 (68.3%) | 420 (66.8%) | | | 289 (31.7%) | 175 (28.6%) | | | 288 (30.6%) | 176 (30.3%) | | |
| **What is your main professional affiliation type?** | | | | | | | | | | | | | |
| Competitive hiring process (generally public) | 801 (52.8) | 456 (51.1%) | 345 (54.8%) | 16.64 | **0.005** | 461 (50.6%) | 340 (55.6%) | 6.15 | **0.013** | 491 (52.2%) | 310 (53.4%) | 17.327 | **0.004** |
| Salaried position | 370 (24.4) | 243 (27.2%) | 127 (20.2%) | | | 252 (27.7%) | 118 (19.3%) | | | 245 (26.0%) | 125 (21.5%) | | |
| Independent professional | 121 (8.0) | 62 (6.9%) | 59 (9.4%) | | | 60 (6.6%) | 61 (10.0%) | | | 60 (6.4%) | 61 (10.5%) | | |
| Cooperative | 48 (3.2) | 27 (3.0%) | 21 (3.3%) | | | 29 (3.2%) | 19 (3.1%) | | | 31 (3.3%) | 17 (2.9%) | | |
| Scholarship recipient/fellow | 95 (6.3) | 63 (7.1%) | 32 (5.1%) | | | 59 (6.5%) | 36 (5.9%) | | | 68 (7.2%) | 27 (4.6%) | | |
| Other | 81 (5.3) | 40 (4.5%) | 41 (6.5%) | | | 47 (5.2%) | 34 (5.6%) | | | 43 (4.6%) | 38 (6.5%) | | |
| **Workplace** | | | | | | | | | | | | | |

(*Continued*)

**Table 1.** (Continued)

| Variable | N (%) | Depression | | | | Anxiety | | | | Stress | | | |
|---|---|---|---|---|---|---|---|---|---|---|---|---|---|
| | | Present N = 893 n (%) | Absent N = 629 n (%) | $X^2$ | p-value | Present N = 911 n (%) | Absent N = 611 n (%) | $X^2$ | p-value | Present N = 941 n (%) | Absent N = 581 n (%) | $X^2$ | p-value |
| Primary health care | 473 (32) | 286 (32.0%) | 187 (29.7%) | 10.74 | **0.013** | 283 (31.1%) | 190 (31.1%) | 1.26 | 0.260 | 296 (31.5%) | 177 (30.5%) | 2.570 | 0.109 |
| Hospital | 563 (38) | 348 (39.0%) | 215 (34.2%) | | | 362 (39.7%) | 201 (32.9%) | | | 369 (39.2%) | 194 (33.4%) | | |
| Other | 395 (26.7) | 205 (22.9%) | 190 (30.2%) | | | 209 (22.9%) | 186 (30.4%) | | | 221 (23.5%) | 174 (29.9%) | | |
| Two professional roles in workplaces | 49 (3.3) | 31 (3.5%) | 18 (2.9%) | | | 34 (3.7%) | 15 (2.5%) | | | 31 (3.3%) | 18 (3.1%) | | |
| **Weekly workload** | | | | | | | | | | | | | |
| 1–10 hours | 46 (3.0) | 26 (2.9%) | 20 (3.2%) | 9.374 | 0.154 | 26 (2.9%) | 20 (3.3%) | 12.58 | **<0.001** | 27 (2.9%) | 19 (3.3%) | 12.71 | **<0.001** |
| 11–20 hours | 64 (4.3) | 36 (4.0%) | 28 (4.5%) | | | 31 (3.4%) | 33 (5.4%) | | | 31 (3.2%) | 33 (5.7%) | | |
| 21–30 hours | 97 (6.5) | 51 (5.7%) | 46 (7.3%) | | | 52 (5.7%) | 45 (7.4%) | | | 58 (6.2%) | 39 (6.7%) | | |
| 31–40 hours | 637 (43) | 356 (39.9%) | 281 (44.7%) | | | 363 (39.8%) | 274 (30.1%) | | | 378 (40.2%) | 259 (44.6%) | | |
| 41–50 hours | 306 (20.6) | 191 (21.4%) | 115 (18.3%) | | | 193 (21.2%) | 113 (12.4%) | | | 192 (20.4%) | 114 (19.6%) | | |
| 51–60 hours | 179 (12.1) | 117 (13.1%) | 62 (9.8%) | | | 123 (13.5%) | 56 (6.1%) | | | 130 (13.8%) | 49 (8.4%) | | |
| >60 hours | 153 (10.3) | 94 (10.5%) | 59 (9.6%) | | | 100 (11.0%) | 53 (5.8%) | | | 104 (11.0%) | 49 (8.4%) | | |

Note: The total number of responses to the variables are not uniform considering *missing*. *Professional category abbreviations*: COREN- Nursing Board; CRM- Board of Medicine; CRO- Board of Dentistry; CRF- Board of Pharmacy CREFITO- Board of Physiotherapy.

for stress [30]. During the first quarter of 2021 in Brazil, 21.5% of adult Brazilians exhibited severe/extreme signs and symptoms of stress, 19.4% anxiety, and 21.5% depression [11], nearly double the rates during the second quarter of 2020 [10].

The situation was even more concerning among healthcare workers; our findings highlight a higher prevalence of stress (61.4%), anxiety (59.7%), and depression (58.7%) compared to the general population [4–6] and other studies involving healthcare workers [31–34].

Symptoms of depression associated with medical training are not surprising, and were observed during the COVID-19 pandemic [35,36]. The conditions that combine to produce this finding among physicians can be explained by low levels of social support, strenuous workload, scarcity of medical equipment, discriminatory experiences, and even violence in the workplace resulting from communication difficulties between doctors and patients' family members [16,33].

Physicians' fears of being infected [36] and even feeling helpless with regard to the services they provide [34] can also justify the association between depression and feelings of insecurity with the organization and structure of services. Physicians are often responsible for final decisions on patient care [37], which may explain why other professional categories were identified with protective factors against depression in this study.

The protective factor associated with marital status has been well described [38]. Single professionals had a higher risk of mental illness symptoms related to stress, anxiety, and depression during social distancing in the pandemic [11]. Married individuals also tend to be healthier [39], possibly due to the positive effects that family stability can have on mental health.

**Table 2. Descriptive analysis and association of variables related to Covid-19 and working conditions with outcomes (N = 1,522), MS/DF, Brazil, 2021.**

| Variable | N (%) | Depression | | | | Anxiety | | | | Stress | | | |
|---|---|---|---|---|---|---|---|---|---|---|---|---|---|
| | | Present N = 893 n (%) | Absent N = 629 n (%) | X² | p-value | Present N = 911 n (%) | Absent N = 611 n (%) | X² | p-value | Present N = 941 n (%) | Absent N = 581 n (%) | X² | p-value |
| **How would you evaluate your physical health, considering your disposition for current personal and professional demands during the pandemic?** | | | | | | | | | | | | | |
| Excellent | 127 (8.5) | 30 (3.43) | 97 (15.65) | 26.24 | <0.001 | 40 (4.47) | 40 (8.06) | 14.13 | <0.001 | 35 (3.79) | 92 (16.14) | 27.61 | <0.001 |
| Good | 577 (38.6) | 252 (28.83) | 325 (52.42) | | | 260 (29.08) | 260 (52.42) | | | 277 (29.98) | 300 (52.63) | | |
| Moderate | 551 (36.9) | 390 (44.62) | 161 (25.97) | | | 396 (44.30) | 155 (31.25) | | | 404 (43.72) | 147 (25.79) | | |
| Poor | 239 (16) | 202 (23.11) | 37 (5.97) | | | 198 (22.15) | 41 (8.27) | | | 208 (22.51) | 31 (5.44) | | |
| **How would you evaluate your mental health, considering your disposition for current personal and professional demands during the pandemic?** | | | | | | | | | | | | | |
| Excellent | 90 (6) | 9 (1.03) | 81 (13.06) | 69.54 | 0.001 | 11 (1.23) | 79 (13.17) | 39.22 | <0.001 | 9 (0.98) | 81 (9.56) | 43.41 | <0.001 |
| Good | 499 (33.4) | 159 (18.21) | 340 (54.84) | | | 172 (19.26) | 327 (54.50) | | | 165 (17.88) | 334 (39.43) | | |
| Moderate | 595 (39.1) | 411 (47.08) | 184 (29.68) | | | 432 (48.38) | 163 (27.17) | | | 456 (49.40) | 139 (16.41) | | |
| Poor | 309 (20.7) | 294 (33.68) | 15 (2.42) | | | 278 (31.13) | 31 (5.17) | | | 293 (31.74) | 293 (34.59) | | |
| **Have you been diagnosed with Covid?** | | | | | | | | | | | | | |
| Yes | 1008 (67.7) | 312 (35.78) | 169 (27.39) | 11.24 | 0.001 | 327 (36.70) | 154 (25.84) | 19.11 | <0.001 | 324 (35.18) | 157 (27.64) | 8.78 | 0.003 |
| No | 481 (32.3) | 560 (64.22) | 448 (72.61) | | | 564 (63.30) | 444 (74.50) | | | 597 (64.82) | 411 (72.36) | | |
| **Outside of work, have you practiced social distancing?** | | | | | | | | | | | | | |
| Yes | 1377 (92.2) | 820 (93.82) | 557 (89.84) | 7.42 | 0.006 | 840 (93.96) | 539 (89.53) | 9.28 | 0.002 | 866 (93.72) | 511 (89.65) | 7.55 | 0.006 |
| No | 117 (7.8) | 54 (6.18) | 63 (10.16) | | | 54 (6.04) | 63 (10.47) | | | 58 (6.28) | 59 (10.35) | | |
| **Do you feel safe with regard to activities involving control, prevention, and care for Covid-19?** | | | | | | | | | | | | | |
| Yes | 410 (27.4) | 179 (20.48) | 231 (37.26) | 26.60 | <0.001 | 192 (21.48) | 218 (36.33) | 24.63 | <0.001 | 189 (20.45) | 221 (38.77) | 30.03 | <0.001 |
| No | 912 (61) | 598 (68.42) | 314 (50.65) | | | 605 (67.67) | 307 (51.17) | | | 632 (68.40) | 280 (49.12) | | |
| I don't know | 172 (11.5) | 97 (11.10) | 75 (12.10) | | | 97 (10.85) | 75 (12.50) | | | 103 (11.15) | 69 (12.11) | | |
| **Do you feel safe with regard to how your work is organized and structured to address the Covid-19 pandemic?** | | | | | | | | | | | | | |

(*Continued*)

**Table 2.** (Continued)

| Variable | N (%) | Depression | | | | Anxiety | | | | Stress | | | |
|---|---|---|---|---|---|---|---|---|---|---|---|---|---|
| | | Present N = 893 n (%) | Absent N = 629 n (%) | X² | p-value | Present N = 911 n (%) | Absent N = 611 n (%) | X² | p-value | Present N = 941 n (%) | Absent N = 581 n (%) | X² | p-value |
| Yes | 387 (26.3) | 164 (19.03) | 223 (36.50) | 18.56 | <0.001 | 175 (19.82) | 212 (35.93) | 6.24 | 0.012 | 325 (35.68) | 123 (21.89) | 2.804 | 0.094 |
| No | 448 (30.4) | 328 (38.05) | 120 (19.64) | | | 317 (35.90) | 131 (22.20) | | | 170 (18.66) | 217 (38.61) | | |
| Partially | 638 (43.3) | 370 (42.92) | 268 (43.86) | | | 391 (44.28) | 247 (41.86) | | | 416 (45.66) | 222 (39.50) | | |
| **How did the pandemic affect your career/work?** | | | | | | | | | | | | | |
| I remained unemployed | 15 (1.2) | 6 (0.83) | 9 (1.67) | | | 6 (0.81) | 9 (1.73) | 6.13 | 0.013 | 6 (0.79) | 9 (1.78) | 2.259 | 0.133 |
| I kept working | 1073 (84.7) | 617 (84.87) | 456 (84.76) | | | 625 (84.01) | 448 (85.99) | | | 642 (84.58) | 431 (85.18) | | |
| I kept working, but from home | 73 (5.8) | 38 (5.23) | 35 (6.51) | | | 42 (5.65) | 31 (5.95) | | | 45 (5.93) | 28 (5.53) | | |
| I started working after the pandemic | 70 (%) | 41 (5.64) | 29 (5.39) | | | 46 (6.18) | 24 (4.61) | | | 43 (5.67) | 27 (5.34) | | |
| I lost my job | 34 (2.7) | 25 (3.44) | 9 (1.67) | | | 25 (3.36) | 9 (1.73) | | | 23 (3.03) | 11 (2.17) | | |
| **Vacation** | | | | | | | | | | | | | |
| Paid | 61 (22.4) | 40 (23.53) | 21 (29.59) | 8.748 | 0.120 | N (%) | N (%) | | | 40 (20.83) | 21 (26.25) | 0.959 | 0.330 |
| Suspended | 211 (77.6) | 130 (76.47) | 81 (79.41) | | | N (%) | N (%) | | | 152 (79.17) | 59 (73.75) | | |
| **Reallocated** | | | | | | | | | | | | | |
| Yes | 124 (8.1) | 85 (9.52) | 39 (6.20) | 4.99 | 0.025 | 86 (9.17) | 38 (6.22) | 4.64 | 0.031 | 95 (10.10) | 29 (4.99) | 11.83 | 0.001 |
| No | 1398 (91.9) | 808 (90.48) | 590 (93.80) | | | 852 (90.83) | 573 (93.78) | | | 846 (89.90) | 552 (95.01) | | |
| **Leave** | | | | | | | | | | | | | |
| No | 1447 (95.1) | 829 (92.83) | 618 (98.25) | 24.72 | <0.001 | 849 (93.19) | 598 (97.87) | 15.70 | <0.001 | 877 (93.20) | 570 (98.11) | 18.37 | <0.001 |
| Leave for Covid | 15 (1.0) | 9 (1.01) | 6 (0.95) | | | 10 (1.10) | 5 (0.82) | | | 10 (1.06) | 5 (0.86) | | |
| Leave for mental health | 30 (2.0) | 29 (3.25) | 1 (0.16) | | | 29 (3.18) | 1 (0.16) | | | 29 (3.08) | 1 (0.17) | | |
| Leave for other reasons | 30 (2.0) | 26 (2.91) | 4 (0.64) | | | 23 (2.52) | 7 (1.15) | | | 25 (2.66) | 5 (0.86) | | |
| **How did the pandemic affect your family income?** | | | | | | | | | | | | | |
| Increased | 159 (10.4) | 96 (11.09) | 63 (10.21) | 6.74 | **0.009** | 60 (7.08) | 60 (10.05) | 4.77 | 0.029 | 100 (10.92) | 59 (10.41) | 6.20 | 0.013 |
| Remained the same | 777 (51.1) | 432 (49.88) | 345 (55.92) | | | 441 (52.07) | 336 (56.28) | | | 455 (49.67) | 322 (56.79) | | |
| Reduced slightly | 365 (24.0) | 206 (23.79) | 159 (25.77) | | | 218 (25.74) | 147 (24.62) | | | 231 (25.22) | 134 (23.63) | | |
| Reduced significantly | 182 (12.3) | 132 (15.24) | 50 (8.10) | | | 128 (15.11) | 54 (9.05) | | | 130 (14.19) | 52 (9.17) | | |

(*Continued*)

**Table 2.** (Continued)

| Variable | N (%) | Depression | | | | Anxiety | | | | Stress | | | |
|---|---|---|---|---|---|---|---|---|---|---|---|---|---|
| | | Present N = 893 n (%) | Absent N = 629 n (%) | $X^2$ | p-value | Present N = 911 n (%) | Absent N = 611 n (%) | $X^2$ | p-value | Present N = 941 n (%) | Absent N = 581 n (%) | $X^2$ | p-value |
| **In 2020, were you receiving psychological and/or psychiatric treatment or follow-up prior to the Covid-19 pandemic?** | | | | | | | | | | | | | |
| Yes | 377 (25.3) | 249 (28.56) | 128 (20.75) | 11.24 | **0.001** | 258 (28.92) | 119 (19.93) | 14.81 | <0.001 | 273 (29.67) | 104 (18.28) | 23.54 | <0.001 |
| No | 1112 (74.7) | 623 (71.44) | 489 (79.25) | | | 634 (71.08) | 478 (80.07) | | | 647 (70.33) | 465 (81.72) | | |
| **Did you seek out some type of psychological and/or psychiatric help or treatment during the pandemic?** | | | | | | | | | | | | | |
| Yes | 550 (37) | 395 (45.30) | 155 (25.20) | 61.62 | **<0.001** | 413 (46.30) | 137 (23.03) | 81.96 | <0.001 | 434 (47.17) | 116 (20.46) | 106.28 | <0.001 |
| No | 937 (63) | 477 (54.70) | 460 (74.80) | | | 479 (53.70) | 458 (76.97) | | | 486 (52.83) | 451 (79.54) | | |

Note: The total number of responses is not uniform, considering *missing*.

Anxiety was associated with the level of training among healthcare workers, and increased in frequency according to years of education. Lower education level was protective against anxiety symptoms [36,40]. Additional education increases understanding of the events related to the COVID-19 pandemic, and the perceptions of these workers vary according to policies and rapid changes in information. Changes in safety and protection guidelines related to infection control and use of personal protective equipment were sudden, often changing several times in the same week [41], which led to emotional overload [42], especially among individuals with higher levels of education.

Healthcare workers who were directly in contact with COVID-19 or had close relatives diagnosed with this disease exhibited higher levels of anxiety, depression, and stress. (30.36%) In the Brazilian general population, individuals who had COVID-19 exhibited higher risk of severe and extreme depression, suggesting that this experience negatively affects psychological state [11]. Healthcare workers who had physical symptoms of the disease were more likely to exhibit anxiety (OR = 2.1, 5% CI 1.36; 3.48). Professionals who developed symptoms had to choose whether to take sick leave or continue working to fill staff shortages in health care, and were afraid of infecting their colleagues and relatives [43]; for this reason, it is plausible that not being infected with the disease was protective against anxiety.

Paradoxically, not seeking psychological or psychiatric help was a protective factor for signs and symptoms of both anxiety and stress. Seeking psychological or psychiatric treatment requires an ongoing investment of time, another obligation for workers who were already overloaded due to the pandemic. Furthermore, it is not uncommon for individuals with moderate anxiety to consider their health status acceptable [44]. Access to mental health support is neglected in public health and mental health policies, which may discourage people from seeking such services. The search for and availability of mental health services during the pandemic should be better explored to understand this phenomenon.

Self-perception of poor mental health was independently associated with the healthcare workers. In the theoretical transactional stress model [45], individuals are agents and not victims of the stress process, suggesting that psychological resources can act as protective agents

**Table 3. Adjusted analysis of the variables associated with depression, anxiety, and stress among healthcare professionals in the center-west region of Brazil in 2021.**

| Variable | Depression | |
| --- | --- | --- |
| | RPaj* (95% CI) | p-value |
| Marital status | | |
| Married | 0.88 (0.79; 0.99) | 0.034 |
| Professional category | | |
| CRM (professional registration with regional medical board) | 3.75 (1.59; 8.85) | 0.002 |
| Occupation (professional activity) | | |
| Management | 0.67 (0.47; 0.95) | 0.027 |
| Nurse | 0.75 (0.58; 0.97) | 0.031 |
| Pharmacist | 0.63 (0.46; 0.85) | 0.003 |
| Do you feel safe about how your work is organized and structured to address the Covid-19 pandemic? | | |
| No | 1.12 (1.03; 1.21) | 0.006 |

| Variable | Anxiety | |
| --- | --- | --- |
| | RPaj*(95% CI) | p-value |
| Education | | |
| Specialization | 0.71 (0.54; 0.94) | 0.019 |
| Master's degree | 0.70 (0.51; 0.95) | 0.026 |
| How would you evaluate your mental health, considering your disposition for current personal and professional demands during the pandemic? | | |
| Poor | 4.63 (2.58; 8.31) | <0.001 |
| Moderate | 4.09 (2.29; 7.28) | < 0.001 |
| Have you been diagnosed with Covid-19? | | |
| No | 0.90 (0.83; 0.98) | 0.034 |
| Did you seek out some type of psychological and/or psychiatric help or treatment during the pandemic? | | |
| No | 0.90 (0.82; 0.99) | 0.034 |

| Variable | Stress | |
| --- | --- | --- |
| | RPaj* (95% CI) | p-value |
| Where do you live? | | |
| Mato Grosso do Sul | 0.91 (0.85; 0.98) | 0.017 |
| Professional category | | |
| CRO (professional registration with regional board of dentistry) | 0.81 (0.68; 0.97) | 0.024 |
| How would you evaluate your mental health, considering your disposition for current personal and professional demands during the pandemic? | | |

*(Continued)*

**Table 3.** (Continued)

| | RPaj | p |
|---|---|---|
| Poor | 6.95 (3.65; 13.23) | <0.001 |
| Moderate | 6.11 (3.22; 11.59) | <0.001 |
| Did you seek out some type of psychological and/or psychiatric help or treatment during the pandemic? | | |
| No | 0.88 (0.82; 0.95) | 0.001 |

*RPaj = prevalence ratio with robust adjustment of variance.

in adverse conditions. Belief in one's own abilities to achieve goals (self-efficacy) and the ability to be flexible in critical situations (resilience) can result in lower or higher degrees of stress among workers [46], depending on their perceptions of the stressor. Strategies for mental health promotion should be directed at healthcare workers, teams, and managers.

A lower prevalence of mental health symptoms was observed in dentistry, a result corroborated in another study [47]. This may reflect the lower number of these professionals who work in hospitals. People who work in areas with high COVID-19 infection rates have reported more severe degrees of all psychological symptoms than other healthcare workers [48].

Living in the state of Mato Grosso do Sul during the pandemic period was a protective factor in our study. The numbers of and trends in accumulated COVID-19 cases varied over time among Brazil's federal units, with the most severe scenario seen in the north and northeast of the country [49]. Within the center-west region, the two states studied (Mato Grosso do Sul and Federal District) did not implement quarantine for the entire population, but rather adopted social distancing measures (suspending events and classes, quarantining risk groups, and introducing partial economic stoppages) [50–52], and the regions are politically and epidemiologically similar. The characteristics and organization of the health services require further investigation to better understand the protective relationship between regions of the country and the psychological impacts identified in healthcare workers.

This study has some limitations that should be addressed. Since it is cross-sectional in nature, there was a limited ability to separate pre-existing and new symptoms, as well as whether mental health in workers was in fact more affected throughout the pandemic, and for these reasons a longitudinal study is required. The application of online questionnaires to assess mental health status adds a non-random selection bias that raises the risk that symptoms may be overestimated. However, this methodological approach was the only ethically acceptable option during the study period. Additionally, evidence suggests that remote online screening results in estimates comparable to face-to-face screening, and these methods in themselves are not a concern [53]. The evaluation was sectional but involved two waves of the COVID-19 pandemic, which could lead to reverse causality, since workers with a history of mental illness symptoms prior to the pandemic may be more likely to present a higher perception of illness risk than those without symptoms of depression, anxiety, or stress.

Given that the COVID-19 pandemic has not been officially declared over, our article remains relevant to the current scenario. It provides an opportunity to assess the present state of the pandemic and explore potential practical strategies for the future. It is important to note that new pandemics are expected to emerge and, as a result of low vaccine coverage worldwide, there is a possibility of reemergence of vaccine-preventable diseases. Hence, further follow-up

studies should be conducted, particularly on the impact of strategies in local contexts, especially in low- and middle-income countries.

Finally, studies suggest that health services should identify and provide ongoing training to verify signs of mental health related illness in healthcare professionals, as well as appropriate coping tools for disaster situations and public emergencies, and should also discuss strategies to alleviate the impact of the ongoing pandemic on mental health in health professionals [54,55]. Considering that different variables were relevant for each type of symptom, these findings can help develop more effective actions to promote and protect mental health among healthcare workers.

## Conclusion

A mental health assessment of healthcare workers in Brazil revealed a worrying prevalence of mental disorders during the COVID-19 pandemic, perceptions of insecurity, and poor self-perception of mental health associated with all symptoms of mental disorders.

Efforts to adopt strategies in health services during the COVID-19 pandemic were not sufficient to protect the mental health of these workers. New interventions adopted to prevent and increase these symptoms are in line with new international mental health agendas.

## Supporting information

**S1 File.**
(PDF)

**S2 File.**
(DOCX)

**S3 File.**
(PDF)

## Author Contributions

**Conceptualization:** Silvia Helena Mendonça de Moraes, Inara Pereira da Cunha, Everton Ferreira Lemos, Lesly Lidiane Ledezma Abastoflor, Maria de Lourdes Oshiro, Rosana Teresinha D. Orio de Athayde Bohrer, Vicente Sarubbi, Jr, Fabrícia Barros de Souza, Débora Dupas Gonçalves do Nascimento, Sandra Maria do Valle Leone de Oliveira.

**Formal analysis:** Silvia Helena Mendonça de Moraes, Inara Pereira da Cunha, Everton Ferreira Lemos, Lesly Lidiane Ledezma Abastoflor, Maria de Lourdes Oshiro, Rosana Teresinha D. Orio de Athayde Bohrer, Vicente Sarubbi, Jr, Fabrícia Barros de Souza, Débora Dupas Gonçalves do Nascimento, Sandra Maria do Valle Leone de Oliveira.

**Investigation:** Silvia Helena Mendonça de Moraes, Inara Pereira da Cunha, Everton Ferreira Lemos, Lesly Lidiane Ledezma Abastoflor, Maria de Lourdes Oshiro, Rosana Teresinha D. Orio de Athayde Bohrer, Vicente Sarubbi, Jr, Fabrícia Barros de Souza, Débora Dupas Gonçalves do Nascimento, Sandra Maria do Valle Leone de Oliveira.

**Methodology:** Silvia Helena Mendonça de Moraes, Inara Pereira da Cunha, Everton Ferreira Lemos, Lesly Lidiane Ledezma Abastoflor, Maria de Lourdes Oshiro, Rosana Teresinha D. Orio de Athayde Bohrer, Vicente Sarubbi, Jr, Fabrícia Barros de Souza, Débora Dupas Gonçalves do Nascimento, Sandra Maria do Valle Leone de Oliveira.

**Writing – original draft:** Silvia Helena Mendonça de Moraes, Inara Pereira da Cunha, Everton Ferreira Lemos, Lesly Lidiane Ledezma Abastoflor, Maria de Lourdes Oshiro, Rosana

Teresinha D. Orio de Athayde Bohrer, Vicente Sarubbi, Jr, Fabrícia Barros de Souza, Débora Dupas Gonçalves do Nascimento, Sandra Maria do Valle Leone de Oliveira.

**Writing – review & editing:** Silvia Helena Mendonça de Moraes, Inara Pereira da Cunha, Everton Ferreira Lemos, Lesly Lidiane Ledezma Abastoflor, Maria de Lourdes Oshiro, Rosana Teresinha D. Orio de Athayde Bohrer, Vicente Sarubbi, Jr, Fabrícia Barros de Souza, Débora Dupas Gonçalves do Nascimento, Sandra Maria do Valle Leone de Oliveira.

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
