## [Decision Letter · Decision Letter 0]

15 Mar 2023

PONE-D-22-24627Self-perception of mental health among Brazilian healthcare workers: a web-based cross-sectional studyPLOS ONE

Dear Dr. Oliveira,

Thank you for submitting your manuscript to PLOS ONE. After careful consideration, we feel that it has merit but does not fully meet PLOS ONE’s publication criteria as it currently stands. Therefore, we invite you to submit a revised version of the manuscript that addresses the points raised during the review process.

We look forward to receiving your revised manuscript.

Kind regards,

Hsin-Yen Yen

Academic Editor

PLOS ONE

Journal Requirements:

2. Please include a caption for figure 1.

3. Please include your tables as part of your main manuscript and remove the individual files. Please note that supplementary tables (should remain/ be uploaded) as separate "supporting information" files.

Reviewers' comments:

Reviewer's Responses to Questions

**Comments to the Author**

1. Is the manuscript technically sound, and do the data support the conclusions?

Reviewer #1: Yes

Reviewer #2: Yes

2. Has the statistical analysis been performed appropriately and rigorously? 

Reviewer #1: Yes

Reviewer #2: Yes

3. Have the authors made all data underlying the findings in their manuscript fully available?

Reviewer #1: Yes

Reviewer #2: Yes

4. Is the manuscript presented in an intelligible fashion and written in standard English?

Reviewer #1: Yes

Reviewer #2: Yes

5. Review Comments to the Author

Reviewer #1: Good Research Paper

1.This is a very important ongoing issue of mental health in healthcare workers worldwide.

2. Findings are consistent with findings from other published studies.

3. The article has been presented in a very concise manner.

4. A standardized tool ( DASS- 21) was used

Issues you may need to consider :

1. The study itself focuses on mental health of healthcare workers during the COVID-19 pandemic, however the title does not reflect this.

2. Should having not sought support for mental health be classified as a protective factor? Have you considered causality in this case? Mental health symptoms cause people to seek help and support. Those that did not seek help are not necessarily protected but simply do not need to seek help because they did not have any mental health problems or symptoms were mild.

3. In the methodology only registered professionals were included in the study , however in the data collection school completion is included. Would you clarify if it is possible to be registered without completing school in Brazil? The general assumption is if someone is registered with a professional body it means they have completed school and have met all the requirements to be a licensed professional.

Reviewer #2: Dear authors,

I read with interest your work. Here are some comments, that I hope that will help you in the publication of your manuscript.

" The repercussions of Covid-19 for mental health among healthcare workers have

been underestimated, and future consequences have not been widely discussed. [7]": I don't think this is quite right, the mental aspect of healthcare workers has been widely addressed since the beginning of pandemic. I strongly recommend to explore literature in this sense and provide a precise framework of the phenomenon. You can check these references if you wish: 10.3390/ijerph19116811, https://doi.org/10.3390/su132413869.

- also you have another problem, that is the timing of the study, in late 2020 to 2021. You should explain in the introduction why your data are still important to be published (current situation? future crisis? etc).

- try to be more precise in stating the aims of the study (when? how? where?)

- the formula in methods needs explanations

- divide Method in subheadings, because in this form is very hard to follow

- it is not clear how the participants were selected, explain better the process

- table 1 needs footnotes for the acronyms

- in the results, please add the information about regression (in the multivariate analysis it was found...). A table including multivariate results may also be useful.

- In discussion you should focus on future perspectives and practical implications to make your reasearch updated.

-

6. PLOS authors have the option to publish the peer review history of their article (what does this mean?). If published, this will include your full peer review and any attached files.

Reviewer #1: No

Reviewer #2: No

---

## [Author Response · Author response to Decision Letter 0]

29 Apr 2023

Campo Grande, April 29, 2023

For PLOS ONE

Revision required [PONE-D-22-24627]

Dear Hsin-Yen Yen

Academic Editor,

We appreciate the evaluation in the paper “Self-perception of mental health among Brazilian healthcare workers: a web-based cross-sectional study”, submitted to Plos One.

We carry out the required revisions in accordance with the reviewers.

Journal Requirements:

Response: A review was carried out in the paper meeting the above criteria.

2. Please include a caption for figure 1.

Response: The caption was included in figure 1, which represents "Flowchart of participants with symptoms of mental disorder according to the location of residence".

3. Please include your tables as part of your main manuscript and remove the individual files. Please note that supplementary tables (should remain/ be uploaded) as separate "supporting information" files.

Response: We have included our tables 1, 2 and 3 at the end of the paper, upon request.

Response: We include Content Copyright Permission Form

Review Comments to the Author

Reviewer 1: 

1.This is a very important ongoing issue of mental health in healthcare workers worldwide.

2. Findings are consistent with findings from other published studies.

3. The article has been presented in a very concise manner.

4. A standardized tool ( DASS- 21) was used

Issues you may need to consider :

1. The study itself focuses on mental health of healthcare workers during the COVID-19 pandemic, however the title does not reflect this.

Response: We adapted the title in order to address concerns about mental health in times of a pandemic. 

Prevalence and Associated Factors of Mental Health Disorders among Brazilian Healthcare Workers during the COVID-19 Pandemic: A Web-Based Cross-Sectional Study"

2. Should having not sought support for mental health be classified as a protective factor? Have you considered causality in this case? Mental health symptoms cause people to seek help and support. Those that did not seek help are not necessarily protected but simply do not need to seek help because they did not have any mental health problems or symptoms were mild.

Response: We appreciate your observation. We agree that cross-sectional studies have limitations when it comes to establishing causality, and therefore, we acknowledge that it is not possible to definitively generate hypotheses based on this type of study. However, we have included a possible explanation in the text, and have also highlighted this limitation in the discussion of the study's limitations.

3. In the methodology only registered professionals were included in the study, however in the data collection school completion is included. Would you clarify if it is possible to be registered without completing school in Brazil? The general assumption is if someone is registered with a professional body it means they have completed school and have met all the requirements to be a licensed professional.

Response: Thank you for bringing up this concern, as it is an important one to address. In our study, we took great care to only include participants who had completed their studies and were linked to a professional body, which is only possible when they have voluntarily joined and provided proof of completion of their studies. Furthermore, as part of the consent process, we requested that participants provide their Class Council number and training time profile, which were validated with the active subscribers of each council.

It should be noted that in Brazil, there are healthcare professionals with technical qualifications and high school education who are recognized and registered in their professional bodies. To ensure that our study only included qualified professionals, we specifically requested the Class Council number from each participant as part of the consent process. This information was later validated with the active subscribers of each council, further ensuring that only qualified professionals were included in the study.

Reviewer 2:

I read with interest your work. Here are some comments, that I hope that will help you in the publication of your manuscript.

" The repercussions of Covid-19 for mental health among healthcare workers have been underestimated, and future consequences have not been widely discussed. [7]": I don't think this is quite right, the mental aspect of healthcare workers has been widely addressed since the beginning of pandemic. I strongly recommend to explore literature in this sense and provide a precise framework of the phenomenon. You can check these references if you wish: 10.3390/ijerph19116811, https://doi.org/10.3390/su132413869.

- also you have another problem, that is the timing of the study, in late 2020 to 2021. You should explain in the introduction why your data are still important to be published (current situation? future crisis? etc).

Response: We thank the reviewer, and agree with the change. We change the text to:

The mental illness of healthcare workers is a complex and multifaceted phenomenon, influenced by individual, interpersonal, organizational, and social factors (Lulli et al., 2021; Grazzini et al., 2022). To protect the mental health of these professionals, it is important to implement protective measures, such as organizational support, effective communication, adequate training, and access to mental health resources, to mitigate risks and promote the resilience of healthcare workers in times of health crisis (Pope et al., 2020).

 Despite the investigation of psychosocial risks and protective measures for the mental health of healthcare workers during the COVID-19 pandemic (Pope et al., 2020; Lulli et al., 2021; Grazzini et al., 2022), further research is still needed. New virus variants are emerging and the pandemic is still evolving, which may have a different impact on the mental health of healthcare professionals (Spoorthy, et al., 2020). In addition, working conditions for healthcare professionals can vary significantly across different countries and regions, and the support and protection measures offered to workers can also vary (Chew, et al., 2021). 

Given that the COVID-19 pandemic has not been officially declared over, our article remains relevant to the current scenario. It provides an opportunity to assess the present state of the pandemic and explore potential practical strategies for the future. It is important to note that new pandemics are expected to emerge and, as a result of low vaccine coverage worldwide, there is a possibility of reemergence of vaccine-preventable diseases. Hence, further follow-up studies should be conducted, particularly on the impact of strategies in local contexts, especially in low- and middle-income countries.

Therefore, it is important to continue studying the mental health of healthcare workers to understand the specific needs of professionals in different contexts and to develop tailored interventions that can meet their needs (Kisely et al., 2020).

- try to be more precise in stating the aims of the study (when? how? where?)

Response: We agree with the remarks in the objective, and more precisely include the information:

The objective of this study was to evaluate the mental health of healthcare Workers in Brazil, estimating the prevalence of mental health disorders, and investigating associated factors, perceptions of safety, and self-perceptions about mental health in times of the COVID-19 pandemic.

- the formula in methods needs explanations; - divide Method in subheadings, because in this form is very hard to follow; - it is not clear how the participants were selected, explain better the process

Response: The materials and methods have been corrected to meet the suggestions listed above. Fixes were met with inclusion in subheadings.

- table 1 needs footnotes for the acronyms

Response: Thanks for the observation in Table 1. The acronyms were included.

Professional category abbreviations: COREN- Nursing Board; CRM- Board of Medicine; CRO- Board of Dentistry; CRF- Board of Pharmacy CREFITO- Board of Physiotherapy.

- in the results, please add the information about regression (in the multivariate analysis it was found...). A table including multivariate results may also be useful.

Response: Thanks for the observation. We would like to clarify that in Table 3, we presented a multivariate analysis using Poisson regression. We included only the variables that were found to be statistically significant in the univariate analysis, as including non-significant variables would not add value to the analysis and could potentially lead to overfitting of the model. 

- In discussion you should focus on future perspectives and practical implications to make your reasearch updated.

Response: We include a paragraph in the discussion

 Given that the COVID-19 pandemic has not been officially declared over, our article remains relevant to the current scenario. It provides an opportunity to assess the present state of the pandemic and explore potential practical strategies for the future. It is important to note that new pandemics are expected to emerge and, as a result of low vaccine coverage worldwide, there is a possibility of reemergence of vaccine-preventable diseases. Hence, further follow-up studies should be conducted, particularly on the impact of strategies in local contexts, especially in low- and middle-income countries.

---

## [Editor Report · Decision Letter 1]

23 May 2023

Prevalence and associated factors of Mental Health Disorders among Brazilian Healthcare Workers in times of the COVID-19 Pandemic: A Web-Based Cross-Sectional Study

PONE-D-22-24627R1

Dear Dr. Oliveira,

We’re pleased to inform you that your manuscript has been judged scientifically suitable for publication and will be formally accepted for publication once it meets all outstanding technical requirements.

Kind regards,

Hsin-Yen Yen

Academic Editor

PLOS ONE

---

## [Editor Report · Acceptance letter]

29 May 2023

PONE-D-22-24627R1 

Prevalence and Associated Factors of Mental Health Disorders among Brazilian Healthcare Workers in times of the COVID-19 Pandemic: A Web-Based Cross-Sectional Study 

Dear Dr. Oliveira:

I'm pleased to inform you that your manuscript has been deemed suitable for publication in PLOS ONE. Congratulations! Your manuscript is now with our production department. 

Kind regards, 

on behalf of

Dr. Hsin-Yen Yen 

Academic Editor

PLOS ONE